# Talin dissociates from RIAM and associates to vinculin sequentially in response to the actomyosin force

Clémence Vigouroux[1], Véronique Henriot[1] & Christophe Le Clainche [1✉]

Cells reinforce adhesion strength and cytoskeleton anchoring in response to the actomyosin force. The mechanical stretching of talin, which exposes cryptic vinculin-binding sites, triggers this process. The binding of RIAM to talin could regulate this mechanism. However, the mechanosensitivity of the talin-RIAM complex has never been tested. It is also not known whether RIAM controls the mechanosensitivity of the talin-vinculin complex. To address these issues, we designed an in vitro microscopy assay with purified proteins in which the actomyosin force controls RIAM and vinculin-binding to talin. We demonstrate that actomyosin triggers RIAM dissociation from several talin domains. Actomyosin also provokes the sequential exchange of RIAM for vinculin on talin. The effect of RIAM on this force-dependent binding of vinculin to talin varies from one talin domain to another. This mechanism could allow talin to biochemically code a wide range of forces by selecting different combinations of partners.

---

[1] Université Paris-Saclay, CEA, CNRS, Institute for Integrative Biology of the Cell (I2BC), 91198 Gif-sur-Yvette, France. ✉email: christophe.leclainche@i2bc.paris-saclay.fr

Mechanical cues govern a variety of biological processes. During migration, cells sense and respond to changes in intracellular and extracellular forces by adapting their shape, dynamics, and adhesion. Focal adhesions (FAs), that play a major role in this mechanosensitive process, are composed of transmembrane integrins that mechanically couple the extracellular matrix (ECM) to the actomyosin cytoskeleton, via actin-binding proteins (ABPs)[1–6]. The mechanosensitivity of FAs allows cells to adapt adhesion strength to the internal force of the actomyosin cytoskeleton and to the physical properties of the ECM[7–13]. However, the biochemical mechanisms that govern FA mechanosensitivity remain largely unexplored.

Several lines of evidence indicate that the force-dependent conformational change of the actin-binding protein talin might be the initial switch that triggers the maturation of short-lived nascent adhesions into stable FAs, which withstand higher adhesion and actomyosin traction force[3,14–16]. Although it is thought that distinct force-dependent conformations of talin select specific binding partners to trigger appropriate mechanical responses, a limited number of these mechanisms have been discovered.

The mechanosensitive talin–vinculin interaction has long been the hallmark of FA maturation[3]. Experiments of mechanical stretching of talin by atomic force microscopy and magnetic tweezers revealed how force exposes one or more of the 11 cryptic vinculin-binding sites (VBSs) located in the 13 helical bundles of the talin rod domain (R1–R13, Fig. 1a)[14,17–20]. Using an in vitro reconstitution strategy, we demonstrated that the actomyosin force is sufficient to stretch talin, allowing its binding to vinculin[21,22]. A series of cellular and biochemical studies showed that the force-dependent formation of the talin–vinculin complex reinforces actin anchoring to FAs[21,23–26]. This reinforcement is further enhanced by the stability of the talin–vinculin complex in which vinculin locks talin in its stretched conformation[20,21].

Less is known about the specific talin-binding partners in nascent adhesions. First observations in cells showed that the Rap1-GTP-interacting adaptor molecule (RIAM) is localized at the leading edge of migrating cells, where it cooperates with talin and Ena/VASP proteins to promote the formation of actin-based membrane protrusions[27,28]. During this early phase, RIAM could bind to the F3 domain of talin head, which disrupts the autoinhibition of talin, allowing it to activate integrins[29,30]. Interestingly, RIAM, which is initially enriched in nascent adhesions, is replaced by vinculin in mature FAs, that are known to experience higher traction force[31,32]. Structural studies showed that RIAM also interacts with exposed residues in the talin helical bundles R2, R3, R8, and R11[33]. In R2, R3, and R8, RIAM-binding sites and VBSs overlap[33–35], suggesting a complex interplay between RIAM, vinculin, and talin.

Altogether these observations suggest that the actomyosin force triggers the mechanosensitive transition between nascent adhesions and FAs by controlling the binding of RIAM and vinculin to talin. However, the mechanosensitivity of the talin–RIAM complex has never been tested. It is also not known whether RIAM-binding to talin controls the mechanosensitive formation of the talin–vinculin complex. Furthermore, several domains of talin that interact with RIAM and vinculin may respond to force differently. To determine the mechanosensitivity, sequence and interdependence of these talin-associated reactions, we designed an in vitro microscopy assay with purified proteins. In this assay, the actomyosin force controls the binding of RIAM and vinculin to a micropatterned surface coated with talin constructs, which contain variable RIAM- and vinculin-binding sites (Fig. 1a).

## Results

### In vitro reconstitution of talin–RIAM complexes.
The stoichiometry of the talin–RIAM complex was a major problem to

start this study. RIAM contains two talin-binding site (TBS1 and TBS2). TBS1 binds to the R2, R3, R8, and R11 helix bundles of the talin rod and to the subdomain F3 of the head[30,33,35] (Fig. 1a). TBS2 has a very low affinity for talin[35]. However, TBS2 could reinforce TBS1 anchoring by binding to a neighboring site if available. Hence, RIAM TBS1–TBS2 interacts with talin R2–R3 with a 2:2 stoichiometry and a higher affinity than the TBS1–R3 complex[33]. Based on this information, we designed a fluorescent mCherry-RIAM 1-306 protein, encompassing TBS1 and TBS2, to visualize the RIAM–talin interaction in microscopy (Fig. 1a, Supplementary Fig. 1a, b). We also designed a series of minimal talin constructs, made of the F2–F3 domains of the head, that anchor the protein to a micropatterned surface[21], followed by an exchangeable cassette containing variable RIAM-binding sites, and the C-terminal actin-binding domain ABD3 (R13) that binds to the actomyosin cytoskeleton (Fig. 1a, Supplementary Fig. 1a, b). We selected the three following RIAM-binding regions for insertion in our minimal talin constructs: R1–R2–R3 that includes the high affinity R2–R3 part, R7–R8 in which the RIAM-binding R8 bundle is inserted in R7, and the single helical bundle R11 (Supplementary Fig. 1a). We found that RIAM binds specifically and with high affinity to disk-shaped micropatterns coated with these three talin constructs (Fig. 1b, c). In contrast, RIAM does not bind to talin F2–F3 alone, whereas it binds to the same construct fused to R1–R2–R3 (Supplementary Fig. 2). This result indicates that, in our experimental setup, talin F3 does not interact with RIAM because it is masked by its interaction with the surface or because, as reported by others, its affinity is too low ($K_d = 32 \,\mu M$)[30].

### Actomyosin-dependent binding of vinculin to talin bundles.
Before testing the effect of actomyosin on talin–RIAM interactions, we first established the mechanosensitivity of our minimal talin constructs using the approach that we developed previously[21,22]. In this method, the force applied to disk-shaped talin-coated micropatterns is sufficient to expose cryptic vinculin-binding sites in talin and recruit fluorescent EGFP-vinculin head (Vh) (Fig. 1a). Here, the force is produced by the association of myosin II with slowly polymerizing actin filaments and applied to talin through its actin-binding domain (R13). Although actomyosin self-organizes transiently, it remains in the disks for at least 1000 s. In this assay, talin R1–R2–R3 shows a strong binding to Vh in the presence of actomyosin, compared with the low constitutive binding measured in the absence of actomyosin (Fig. 2a, b, e, Supplementary Movie 1). Using a procedure that we have already validated for full-length talin[21], we confirmed that the actomyosin-dependent increase in Vh–talin interaction depends on myosin II and not on the bundling and polymerization of the actin network, even for a minimal talin like R1–R2–R3 (Supplementary Fig. 3a, b). We also observed the actomyosin-dependent binding of Vh to talin R11 (Fig. 2c, f, Supplementary Movie 2). R1–R2–R3 recruits more Vh than R11 because it contains five VBSs, whereas R11 contains only one (Supplementary Fig. 1a). However, talin R7–R8 does not bind to Vh in the presence of the same concentration of actin and myosin used to stimulate R1–R2–R3 and R11 (Fig. 2d, g, Supplementary Movie 3). The fact that R7–R8 does not bind Vh at all in the presence of actomyosin also rules out the mechanical exposure of the single VBS of ABD3 in our series of three constructs (Fig. 2d, g). Altogether, our data showed that the actomyosin force is efficiently transmitted to talin through ABD3 to stretch R1–R2–R3 and R11 but not R7–R8.

### Actomyosin-dependent dissociation of RIAM from talin.
To test the mechanosensitivity of the three talin–RIAM complexes,

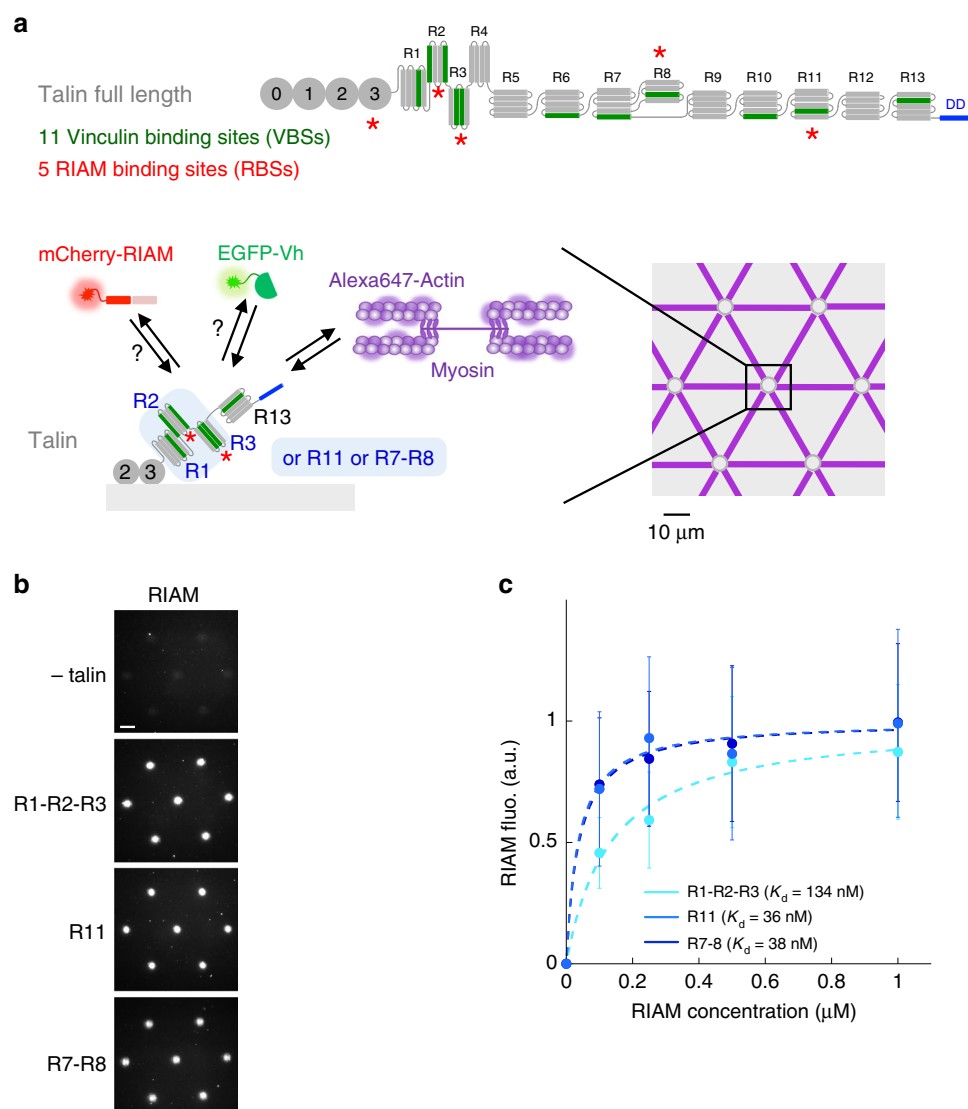

**Fig. 1 In vitro reconstitution of talin–RIAM complexes. a** Top panel: organization of full-length talin featuring RIAM- and vinculin-binding sites. The vinculin-binding sites (VBSs) are the dark green helices. RIAM binds to the R2, R3, R8, and R11 domains of talin. R13 is the C-terminal actin-binding domain (ABD) of talin. Bottom panels: basic principle of the in vitro microscopy assay. Alexa647-labeled actin and myosin II self-assemble to apply force to talin R1–R2–R3, R11, or R7–R8 immobilized in micropatterns, which controls the binding of EGFP-vinculin head (EGFP-Vh) and mCherry-RIAM. Talin is represented as a monomer for convenience but it contains a dimerization domain (DD). **b** Representative images of the fluorescence of mCherry-RIAM 1-306 (1 μM) in non-coated control disks and disks coated with 1 μM talin R1–R2–R3 or R11, or R7–R8. Scale bar = 10 μm. This experiment was repeated three times independently with the same results. **c** Binding of RIAM to disks coated with talin R1–R2–R3, R11, and R7–R8. Conditions: 0–1 μM mCherry-RIAM 1-306, 1 μM of talin during the coating step. Data are mean ± SD. $n = 150$ disks all points, except (R1–R2–R3 + 0.5 μM RIAM) $n = 137$ disks. Source data are provided as a Source data file.

we measured the variation in fluorescence of mCherry-RIAM in talin-coated disks submitted to the force of the actomyosin cytoskeleton. Here, the delay of actomyosin accumulation in the disks makes the onset of force application easy to correlate with force-dependent events. The time lapses and the kymographs revealed that RIAM starts to dissociate from talin R1–R2–R3-coated disks as soon as actomyosin accumulates (Fig. 3a, b, e, Supplementary Movie 4). At the end of the kinetics, 50% of RIAM is dissociated from talin (Fig. 3e). As a control, we also showed that the fluorescence of RIAM bound to talin R1–R2–R3 is stable in the absence of actomyosin (Fig. 3a, b, e, Supplementary Movie 4). Like the actomyosin-dependent binding of Vh to talin constructs, the actomyosin-dependent dissociation of RIAM from this minimal talin does not require

the polymerization of actin, nor the formation of actomyosin bundles, and depends on the presence of myosin II in the assay (Supplementary Fig. 3c, d). This result demonstrates the mechanosensitivity of the talin–RIAM interaction. We used the same method to test the mechanosensitivity of the other two RIAM-binding sites of talin. Similarly, we observed that the actomyosin force triggers the efficient dissociation of RIAM from talin R11 (Fig. 3c, f, Supplementary Movie 5). However, we found that the actomyosin force only provokes a mild dissociation of RIAM from talin R7–R8 in the presence of the same concentration of actin and myosin used to stimulate R1–R2–R3 and R11 (Fig. 3d, g, Supplementary Movie 6). This result is in agreement with the weak mechanosensitivty of talin R7–R8 observed in Fig. 2g.

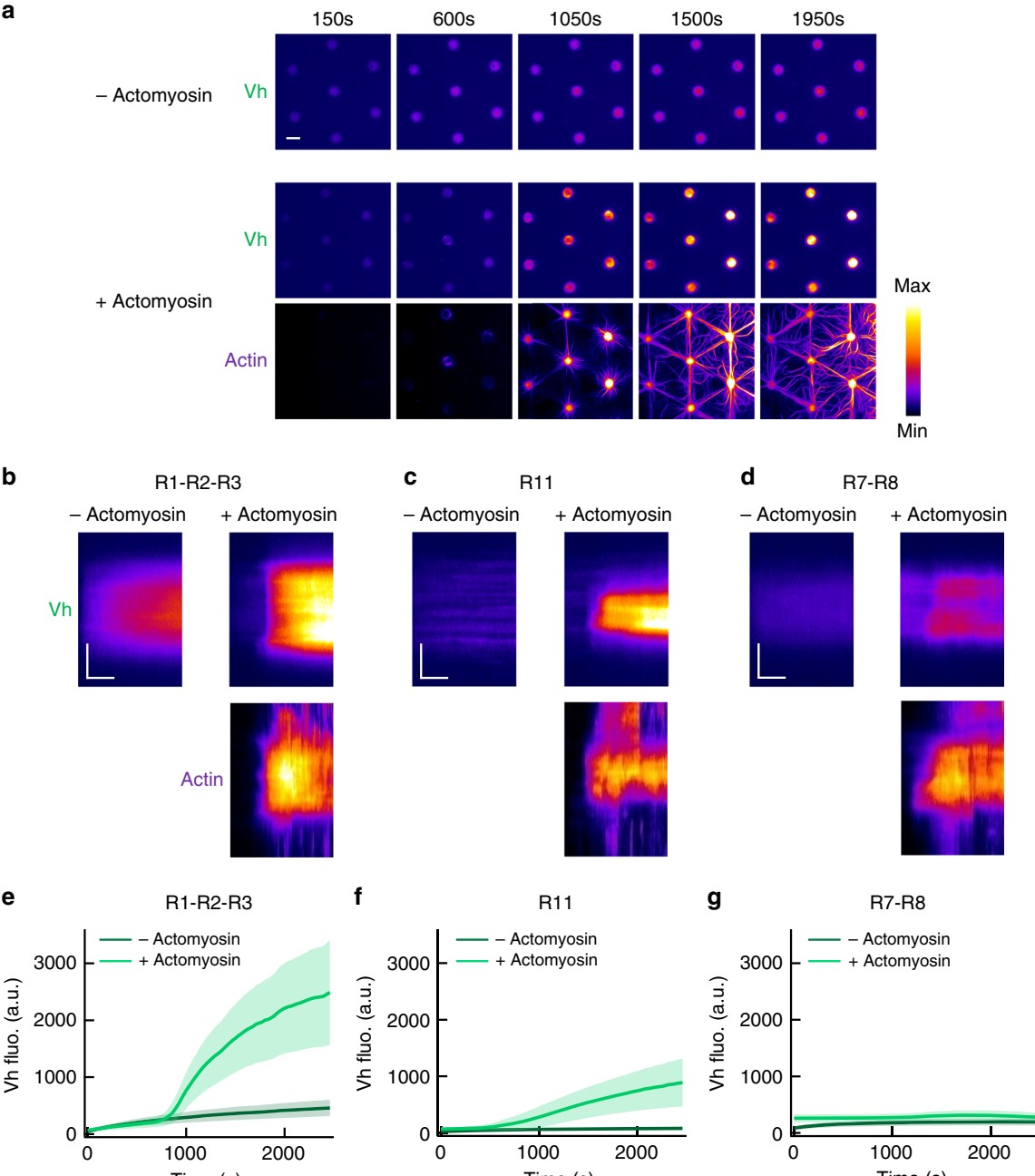

**Fig. 2 Talin domains bind to vinculin differently in response to the actomyosin force. a** Time lapses showing the recruitment of Vh to talin R1–R2–R3-coated disks in the absence (top) or presence of actomyosin (Vh is shown on the middle and actin on the bottom panel). This experiment was repeated five times independently with the same results. **b**-**d** Kymographs of EGFP-Vh (top) and actin (bottom) along a cross-section of a disk coated with talin R1–R2–R3 (**b**) or R11 (**c**) or R7–R8 (**d**) in the absence (left) or presence (right) of actomyosin. Conditions: 100 nM EGFP-Vh, 2.4 μM actin (2% Alexa594-labeled), 50 nM myosin, and 1 μM talin during the coating step. The images are color coded using the fire LUT of ImageJ. Scale bar in time lapses = 10 μm. In kymographs, horizontal bar = 500 s, vertical bar = 5 μm. **e**-**g** Kinetics of the mean fluorescence of EGFP-Vh corresponding to the conditions described in (**b**-**d**). Data are mean ± SD. **e** $n = 80$ (−actomyosin), $n = 60$ disks (+actomyosin). **f** $n = 60$ (−actomyosin), $n = 75$ disks (+actomyosin). **g** $n = 80$ (−actomyosin), $n = 76$ disks (+actomyosin). Source data are provided as a Source data file. See Supplementary Movie 1, Supplementary Movie 2, and Supplementary Movie 3.

**Actomyosin-induced exchange of RIAM for vinculin on talin.** To determine the relationship between the mechanosensitive talin–RIAM and talin–vinculin interactions observed in Fig. 2 and Fig. 3, we compared the kinetics of RIAM, Vh, and actin in disks coated with talin R1–R2–R3 and talin R11 in our assay. The time lapses and the kymographs revealed that actomyosin accumulation is associated with the concomitant RIAM dissociation and Vh association in disks coated with talin R1–R2–R3 (Fig. 4a, b, d, Supplementary Movie 7) and talin R11 (Fig. 4c, f, Supplementary Movie 8). The kinetics halftimes indicate a clear sequence in which actomyosin accumulation is followed by RIAM dissociation and Vh association (Fig. 4d, f). Our observations are

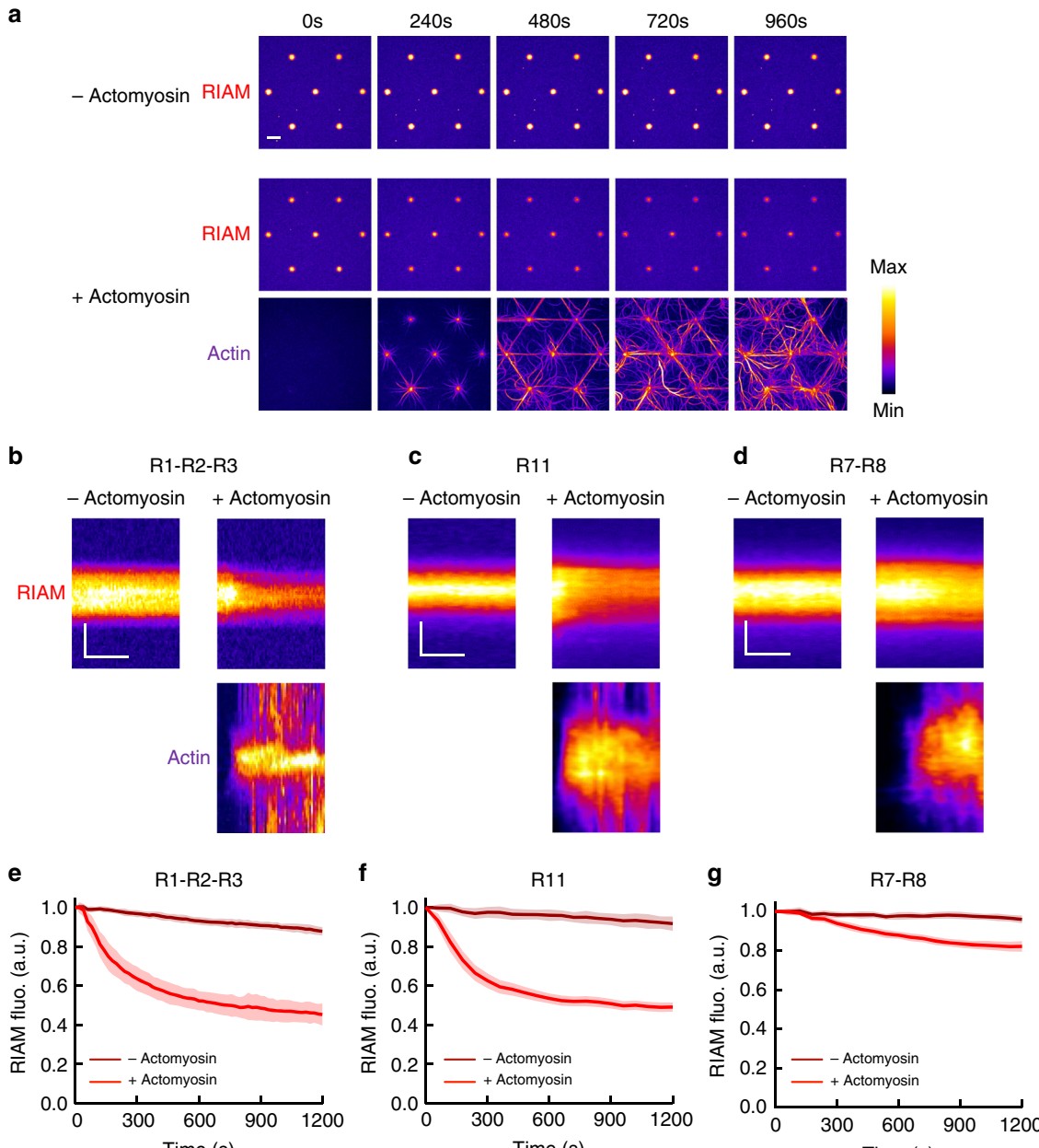

**Fig. 3 The actomyosin force provokes RIAM dissociation from several talin domains. a** Time lapses showing the binding of RIAM 1-306 to talin R1–R2–R3-coated disks in the absence (top) or presence of actomyosin (RIAM is shown on the middle and actin on the bottom panel). This experiment was repeated 11 times independently with the same results. **b-d** Kymographs of mCherry-RIAM 1-306 (top) and actin (bottom) along a cross-section of a disk coated with talin R1-R2-R3 (**b**) R11 (**c**) or R7-R8 (**d**) in the absence (left) or presence (right) of actomyosin. For better comparison, the fluorescence of mCherry-RIAM 1-306 in kymographs was normalized as the maximal fluorescence in the R11- and R7–R8-coated disks. Conditions: 100 nM mCherry-RIAM 1-306, 2.4 μM actin (1% Alexa647-labeled for R1-R2-R3, 2% Alexa488 for R11 and R7-R8), 50 nM myosin, 1 μM talin during the coating step. The images are color coded using the fire LUT of ImageJ. Scale bar in time lapses = 10 μm. In kymographs, horizontal bar = 500 s, vertical bar = 5 μm. **e-g** Kinetics of the mean fluorescence of mCherry-RIAM 1-306 corresponding to the conditions described in (**b-d**). Data are mean ± SD. **e** $n = 63$ (−actomyosin), $n = 62$ disks (+actomyosin). **f** $n = 50$ (−actomyosin), $n = 60$ disks (+actomyosin). **g** $n = 49$ (−actomyosin), $n = 60$ disks (+actomyosin). Data were first normalized to 1 as the maximal mCherry-RIAM 1-306 fluorescence and synchronized using this maximal value as $t_0$ before being averaged. Source data are provided as a Source data file. See Supplementary Movie 4, Supplementary Movie 5, and Supplementary Movie 6.

consistent with a mechanism in which talin switches from a RIAM-specific conformation to a vinculin-specific one. Alternatively, actomyosin could trigger a competition between RIAM and Vh for talin. However, Vh does not affect the dissociation rate of RIAM from talin R1–R2–R3 nor from talin R11 (Fig. 4e, g). The fact that Vh binding to talin follows RIAM dissociation, without influencing it, rules out a direct competition mechanism

and demonstrates that talin behaves as a force-dependent conformational switch.

**Effect of RIAM on the force-induced vinculin–talin complex.** Although Vh does not affect RIAM dissociation (Fig. 4e, g), RIAM could affect talin stretching and its subsequent binding to

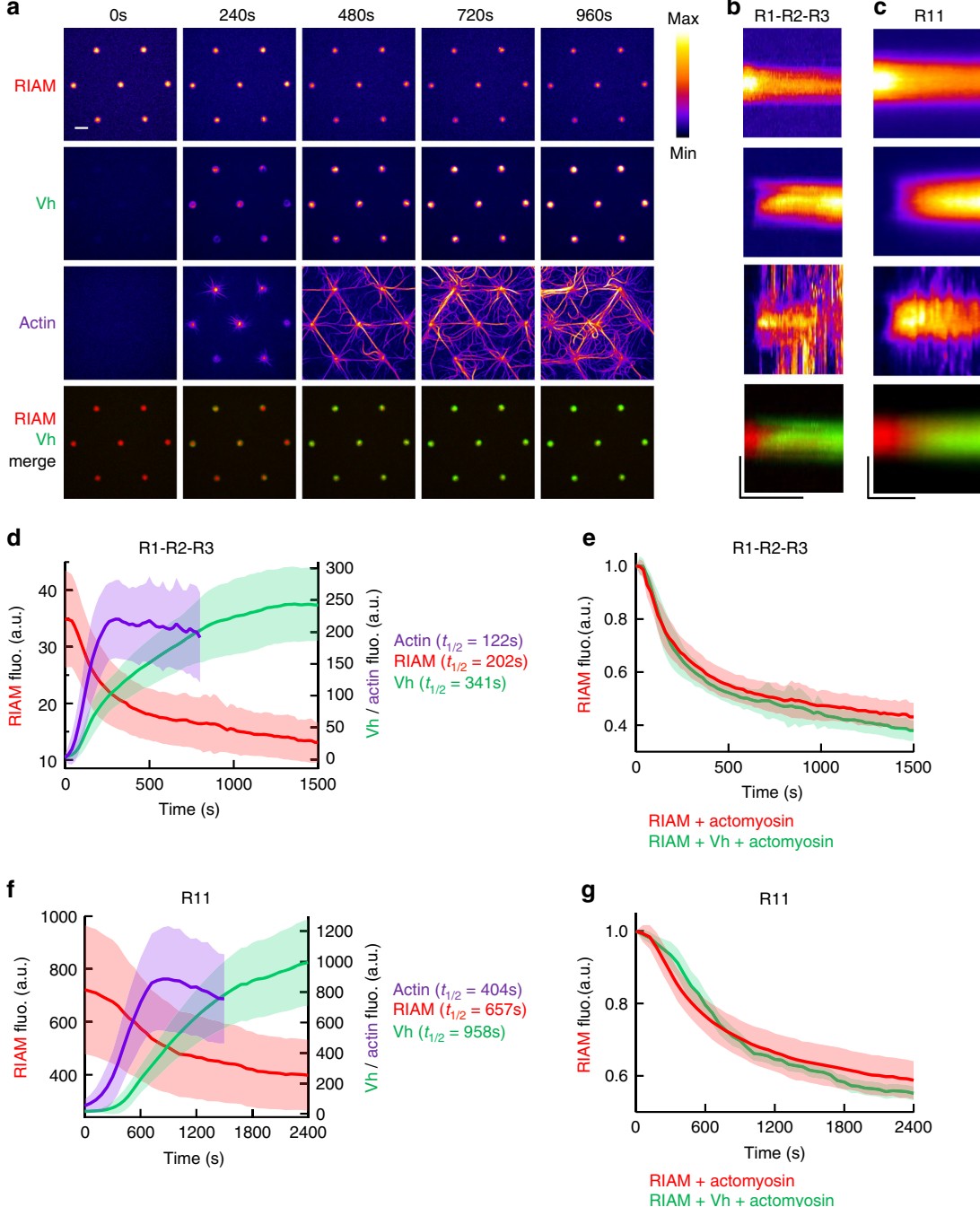

**Fig. 4 The actomyosin force provokes the sequential exchange of RIAM for vinculin on talin. a** Time lapses showing the concomitant dissociation of RIAM 1-306, association of Vh, accumulation of actomyosin, and a Vh/RIAM merge in the same disks coated with talin R1–R2–R3. This experiment was repeated 4 times independently with the same results. **b, c** From top to bottom: kymographs of RIAM, Vh, actin, and Vh/RIAM merge along the cross-section of a disk coated with talin R1–R2–R3 (**b**) or R11 (**c**). Conditions: 100 nM mCherry-RIAM, 100 nM EGFP-Vh, 2.4 μM actin (1% Alexa647-labeled), 50 nM myosin, and 1 μM talin during the coating step. The images are color coded using the fire LUT of ImageJ. Scale bar in time lapses = 10 μm. In kymographs, horizontal bar = 1000 s, vertical bar = 5 μm. **d, f** Kinetics of the mean fluorescence of mCherry-RIAM 1-306, EGFP-Vh and Alexa647-labeled actin in disks coated with talin R1–R2–R3 (**d**) and R11 (**f**) corresponding to the conditions described in (**b**) and (**c**). Actin fluorescence is multiplied by 3. Data are mean ± SD. $n = 63$ (**d**), $n = 59$ disks (**f**). **e** Kinetics of mCherry-RIAM 1-306 dissociation from disks coated with talin R1–R2–R3 in the presence of actomyosin with or without 100 nM EGFP-Vh as described in Fig. 3b and **b** respectively. $n = 62$ (RIAM + actomyosin), $n = 63$ disks (RIAM + Vh + actomyosin). **g** Kinetics of mCherry-RIAM 1-306 dissociation from disks coated with talin R11 in the presence of actomyosin with or without 100 nM EGFP-Vh as described in Fig. 3c and **c** respectively. $n = 59$ disks. **e, g** Data are mean ± SD. Data were first normalized to 1 as the maximal mCherry-RIAM 1-306 fluorescence and synchronized using this maximal value as $t_0$ before being averaged. Source data are provided as a Source data file. See Supplementary Movie 7 and Supplementary Movie 8.

Vh. We therefore measured the effect of a high concentration of RIAM on the actomyosin-dependent binding of Vh to talin. A weak interaction between Vh and RIAM has been reported ($K_d = 5\,\mu M$) and could affect our interpretations[33]. However, this weak interaction cannot compete with the high affinity binding of Vh for a construct of talin corresponding to R1–R2–R3 in which one VBS (helix 12) is exposed ($K_d = 140\,nM$)[36]. Similarly, this weak interaction between Vh and RIAM is unlikely to affect the high affinity binding of RIAM 1-306 to our talin R1–R2–R3 and talin R11 ($K_d = 36\,nM$ and $134\,nM$ respectively, Fig. 1c). In addition, we showed that Vh does not affect the constitutive binding of RIAM to talin R1–R2–R3 in the absence of actomyosin (Supplementary Fig. 4), demonstrating that RIAM and Vh do not sequester each other through direct binding in our experimental conditions. After this clarification, we showed that the actomyosin-dependent binding of Vh to talin R1–R2–R3 is severely impaired by a high concentration of RIAM, compared with the control without RIAM (Fig. 5a, b, Supplementary Movie 9). Kinetic analysis and steady-state measurement confirmed that the final level of Vh bound to stretched talin is lower in the presence of RIAM (Fig. 5d, f), suggesting that the talin–RIAM complex requires a higher force for unfolding than talin alone. Our observations demonstrate that RIAM protects talin R2–R3 bundles from mechanical unfolding, which implies that the force-dependent dissociation of RIAM is a prerequisite for the exposure of VBSs in R2–R3. Surprisingly, RIAM does not affect the actomyosin-dependent binding of Vh to talin R11, at the concentration tested for talin R1–R2–R3 ($0.5\,\mu M$) and at a higher concentration ($3\,\mu M$) (Fig. 5c, e, g, Supplementary Movie 10). Therefore, unlike talin R1–R2–R3, the dissociation of RIAM from talin R11 is not a prerequisite for the force-dependent binding of vinculin.

## Discussion

Our in vitro reconstitution demonstrates that talin dissociates from RIAM and associates to vinculin sequentially in response to the actomyosin force. The force-dependent dissociation of the talin–RIAM interaction is one of the rare elementary mechanosensitive reactions of adhesion complexes which has been discovered. This process could control the mechanosensitive maturation of FAs (Fig. 6a).

We found that RIAM protects R2–R3 against stretching and its dissociation is a prerequisite for vinculin binding. Such a bistable mechanism is the desirable behavior for a mechanosensitive switch involved in a cellular decision-making process. This behavior likely results from the overlap between the RIAM-binding sites and VBSs in R2 and R3[33]. R1–R2–R3 is a key mechanosensitive part of talin, since a construct, almost identical to our talin R1–R2–R3, can rescue spreading, polarization and migration of talin null cells[37]. In this region of talin, R3 has a critical role, since mutations that destabilize this bundle affect ECM sensing[38]. The stretching of single molecules clearly revealed that R3 is the weakest bundle of the R1–R2–R3 region and should unfold first in response to force[19,20]. The inhibition of the mechanical exposure of VBSs in talin R1–R2–R3 by RIAM could therefore result from the stabilization of R3.

Our study reveals that talin R11 binds to vinculin in response to force. In contrast with talin R1–R2–R3, the mechanosensitive binding of R11 to vinculin is not affected by RIAM. Although the structure of the R11–RIAM complex is not known, our data suggest that the binding sites for RIAM and vinculin do not overlap in R11. Talin R11 can therefore dissociate from RIAM and associate to vinculin independently. Interestingly, simulations predicted that the first and last helices of the mechanically-stretched R11 should unfold first, leaving a 3-helix intermediate

bundle containing a cryptic VBS[39]. Whether RIAM dissociates after the detachment of these first and last helices of R11 remains to be determined.

The different effects of RIAM on talin R1–R2–R3 and R11 imply that RIAM increases the threshold force for the exposure of the VBSs located in R2–R3 but does not affect the exposure of the single VBS of R11. However, the sequential exchange of RIAM for vinculin on talin R11 implies that the force required to dissociate RIAM is lower than the one required to expose the single VBS of R11 (Fig. 6b).

R7–R8 is difficult to stretch in conditions that provoke both RIAM dissociation and vinculin association for R1–R2–R3 and R11 (Fig. 6b). In response to force, Vh does not bind to the two VBSs located in R7 and R8 for several possible reasons. The single VBS of R8 is probably not exposed because R8 is protected from unfolding by its insertion in R7. The single VBS of R7 could remain stably attached to R7 in response to force or, if exposed, its affinity for Vh is low as previously reported[40]. The weak dissociation of RIAM and the total absence of vinculin association suggest that the RIAM-binding site is disrupted at a level of force that is not sufficient to expose the single VBS of R8. Altogether, our observations are in agreement with a previous report showing that unfolding of isolated R8 occurs at a force of 5 pN, whereas unfolding of R7–R8 occurs at 15 pN, demonstrating that R8 is mechanically protected by its insertion in R7[18]. Mutations that prevent the mechanical stretching of R7 favor signaling cascades downstream of R8, revealing the importance of this mechanosensitive reaction controlled by a high threshold force[41].

The mechanosensitivity of the domains of talin is influenced by their position relative to the ABDs in the full-length protein. By placing FRET sensors at different positions along talin, Ringer et al. revealed an intramolecular tension gradient characterized by high forces between the head and ABD2 and lower forces between ABD2 and ABD3[42]. However, we showed that a construct composed of R1–R2–R3 fused to ABD2 is not stretched compared with the same construct where ABD2 is replaced by ABD3 (Supplementary Fig. 5). This result can be easily explained by the fact that ABD2 is masked by R3 in talin[23]. The traction force applied to ABD3 would unfold R3, leading to the exposure of ABD2. The fact that R8 is the C-terminal part of ABD2 allows alternative interpretations of the weak mechanosensitivity of R7–R8[23]. Indeed, we cannot exclude that the binding of actin filaments stabilizes R8 and prevents RIAM dissociation and vinculin association. Alternatively, if actomyosin generates a pulling force on R13 (ABD3) and R7–R8 (ABD2) simultaneously, the apparent tension between R13 and R7–R8 could be reduced, as suggested by FRET measurement in cells[42], leading to weak dissociation of RIAM and association of vinculin.

The structure of talin reveals auto-inhibitory contacts between F3 and R9, and also F2 and R12[43,44]. RIAM binding to the talin rod does not require the disruption of the F3–R9 autoinhibition[35]. In contrast, the release of the F3-R9 intramolecular contact appears critical to initiate talin–vinculin interaction independently of force application in nascent adhesion[45]. However, this talin conformation can be further stretched to recruit more vinculin in FAs. Whether the disruption of the talin auto-inhibitory contacts facilitates the stretching of individual bundles to dissociate RIAM and bind vinculin remains to be determined. It would also be interesting to determine whether the recently reported autoinhibition of RIAM controls its interaction with talin[46].

Vinculin autoinhibition influences the mechanosensitivity of the talin–vinculin complex. Several biochemical, structural, and cellular studies compared the recruitment of the constitutively active Vh and the autoinhibited full-length vinculin (VFL) in FAs, leading to apparent discrepancies. In cells, Vh remains associated

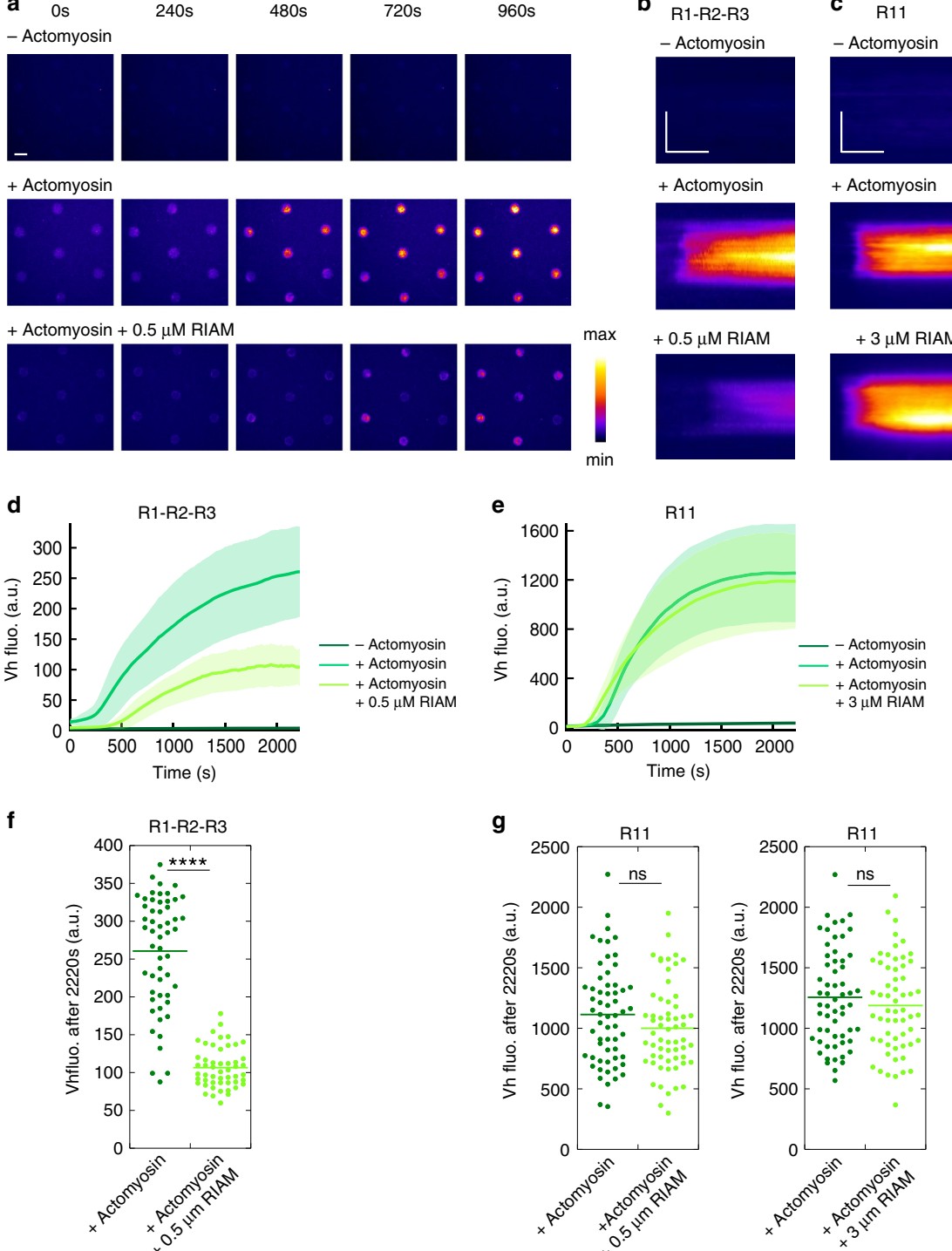

**Fig. 5 RIAM inhibits the actomyosin-dependent binding of vinculin to talin R1–R2–R3 but not to R11. a** Time lapse showing the recruitment of Vh in disks coated with talin R1–R2–R3 in the absence of actomyosin (top), presence of actomyosin (middle), and presence of actomyosin and RIAM (bottom). This experiment was repeated twice independently with the same results. **b**, **c** Kymographs of EGFP-Vh along a cross-section of a disk coated with talin R1–R2–R3 (**b**) and R11 (**c**). Conditions: 100 nM EGFP-Vh, 2.4 µM actin, 50 nM myosin, 500 nM (**b**) or 3 µM (**c**) mCherry-RIAM, 1 µM talin during the coating step. The images are color coded using the fire LUT of ImageJ. Scale bar in time lapses = 10 µm. In kymographs, horizontal bar = 1000 s, vertical bar = 5 µm. **d**, **e** Kinetics of the mean fluorescence of EGFP-Vh corresponding to the conditions described in (**b**, **c**). Data are mean ± SD. **d** $n = 54$ (−actomyosin) and (+actomyosin), $n = 51$ disks (+actomyosin + 0.5 µM RIAM). **e** $n = 59$ disks. **f**, **g** Steady-state binding of Vh (2220 s after sealing the chamber) in disks coated with talin R1–R2–R3 (**f**) or R11 (**g**) in the absence and presence of RIAM. **f**, **g** Same conditions as in (**b**, **c**). Each data point represents the mean fluorescence of Vh in one disk. The bar shows the mean. **f** $n = 54$ (+actomyosin), $n = 51$ disks (+actomyosin + 0.5 µM RIAM). A significant difference was found using a two-tailed $t$ test ($P = 3.95 \times 10^{-22}$). **g** Left panel: $n = 60$. No significant difference was found using a two-tailed $t$ test ($P = 0.1126$). Right panel: $n = 59$ disks. No significant difference was found using a two-tailed $t$ test ($P = 0.3575$). ****$P < 0.0001$ using a two-tailed $t$ test; ns nonsignificant. Source data are provided as a Source data file. See Supplementary Movie 9 and Supplementary Movie 10.

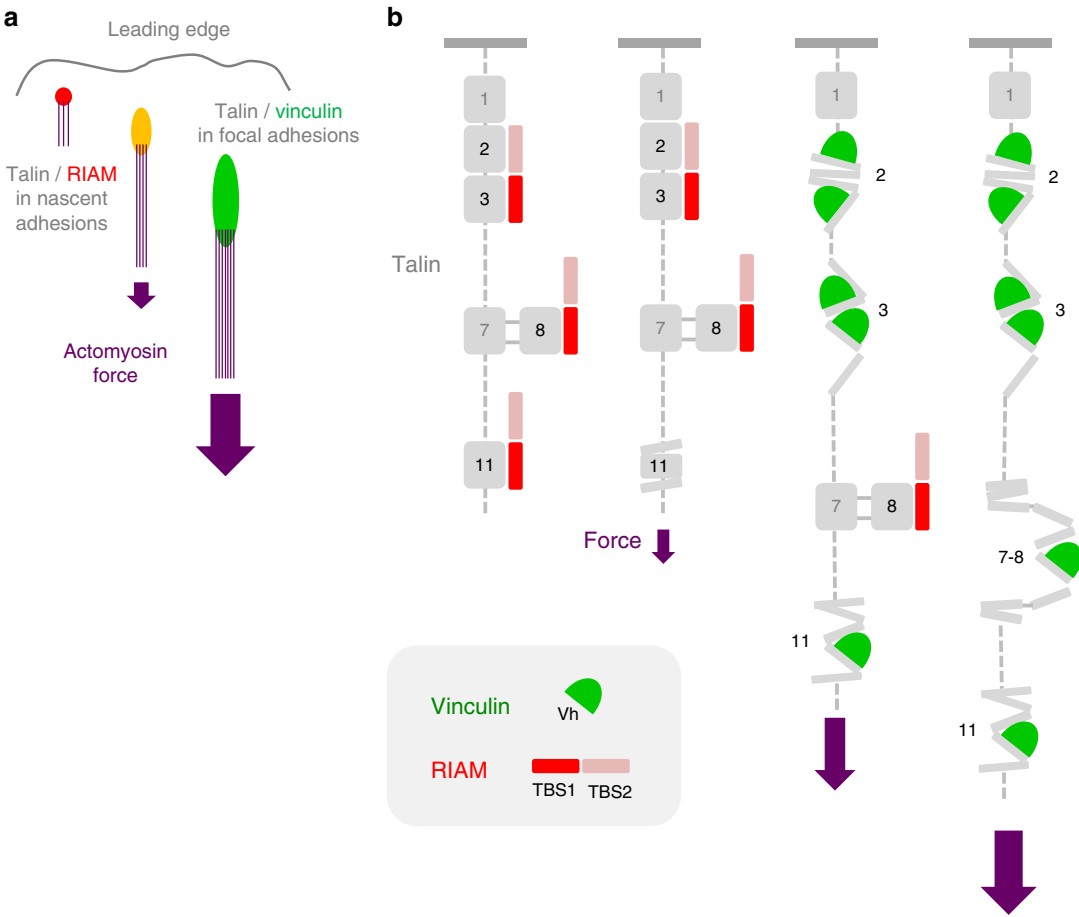

**Fig. 6 Model for the actomyosin-dependent binding of RIAM and vinculin to talin. a** Scheme illustrating that RIAM, which is initially enriched in nascent adhesions, is replaced by vinculin in mature FAs in response to the force exerted by the actomyosin stress fibers. **b** Model describing how talin dissociates from RIAM and associates to vinculin sequentially in response to the actomyosin force.

to talin in FAs after myosin inhibition by blebbistatin, whereas VFL dissociates[24,47,48]. The recruitment of VFL to FAs is restored by cell stretching, demonstrating the force-dependence of the talin–vinculin interaction, whereas Vh binding is not increased[24]. The slow dissociation of Vh from talin after force release, observed in vitro[20,21], could explain the slow dissociation of Vh in cells after blebbistatin treatment, whereas the fast dissociation of VFL could result from the reassociation of the tail to the head of vinculin in the absence of actomyosin. Indeed, actomyosin force acts on vinculin to maintain the active open conformation of vinculin[48,49]. The saturation of talin by Vh would explain why talin does not recruit more Vh after cell stretching. Thus, in vitro, and probably in cells, the mechanosensitivity of the talin–Vh interaction depends on Vh concentration. Because Vh is not autoinhibited like VFL, it binds to partially exposed VBSs in the least stable helical bundles of non-stretched talin, provided that Vh concentration is high enough[21]. At the low concentration of Vh used in our past and present in vitro studies (22–100 nM), Vh mimics VFL by displaying a very weak constitutive binding, which is greatly enhanced by the application of actomyosin force to talin (Fig. 2e).

Talin bundles have several binding partners other than RIAM and vinculin. Interestingly, talin R7 interacts with KANK and R8 interacts with both DLC and paxillin[33,41,50,51]. The catch-to-slip bond switching behavior of the R7–KANK complex is thought to control KANK recruitment at FAs[52]. The mechanical unfolding of R8 should provoke the dissociation of DLC from talin, leading to upregulation of actomyosin contractility, and acceleration of

cell migration[41]. Interestingly, R11 interacts with the β-subunit of integrins. However, it is not known whether R11, like the F3 subdomain of the head, promotes the inside–out activation of integrins and whether this activity is influenced by the force-dependent dissociation of RIAM and association of vinculin. This high number of combinations of talin partners, associated with specific mechanically-stretched talin conformations, provides the cell with a precise means of informing itself about variations in intracellular and extracellular forces. Conversely, the binding of talin partners, as exemplified by the present study on RIAM, can modify the mechanosentivity of talin bundles differently, leading to a change in the hierarchy of their response to force.

## Methods

**cDNA constructs.** All talin constructs are derived from a cDNA encoding for human talin-1 containing a C-terminal His$_6$ tag. Talin R1–R2–R3, corresponding to F2–F3–R1–R2–R3–R13, was cloned into a pETM plasmid with an N-terminal StrepTagII and a C-terminal His$_6$ tag. This construct was made in three steps. First the R2–R3 fragment was PCR amplified using primers 1 and 2 and cloned into the KpnI/BamHI sites of pETM, leading to the intermediate plasmid pETM–R2–R3. R13 was PCR amplified using primers 3 and 4 and cloned into the BamHI/EcoRI sites of pETM–R2–R3, leading to the intermediate plasmid pETM–R2–R3–R13. Finally, F2–F3–R1–R2–R3 was PCR amplified using primers 5 and 6 and cloned into the KpnI/NcoI sites of the pETM–R2–R3–R13, leading to pETM–F2–F3–R1–R2–R3–R13. Talin R11, corresponding to F2–F3–R11–R13 and talin R7–R8, corresponding to talin F2–F3–R7–R8–R13 were cloned into a pET-29a(+) plasmid with an N-terminal StrepTagII and a C-terminal His$_6$ tag. Talin R7–R8 and R11 have been synthesized by Genscript. The cDNAs encoding for talin F2–F3 (talin 196–405), talin F2–F3–R1–R2–R3 (talin 196–911), and talin R1–R8 (talin 196–1659) were PCR amplified using the primer pairs 7–8, 7–9, 7–10 and cloned into the BamH/XhoI, BamHI/EcoRI, and BamHI/EcoRI sites, respectively, of a

pGEX6P1 plasmid (GE Healthcare) with an N-terminal GST tag and a C-terminal His$_6$ tag. Our constructs do not include the F0 and F1 subdomains of the head (talin 1–195) because this part of talin reduces the expression quality in our hands and is not involved in the binding of RIAM and vinculin. A preexisting cDNA encoding for human vinculin 1–851, corresponding to Vh, was cloned into the SalI/NotI sites of a homemade pGEX-6P1-EGFP plasmid. The cDNA encoding for mouse RIAM 1-306 was PCR amplified using primers 11 and 12, and cloned into the BamHI/XhoI sites of a homemade pGEX-6P2-mCherry plasmid with a C-terminal His$_6$ tag. Primers used for cloning the DNA constructs in this study are listed in Supplementary Table 1.

**Protein purification**. All the recombinant proteins were expressed in Escherichia coli (BL21 DE3, Invitrogen). After transformation, bacteria were grown in 4–12 l of LB medium containing 0.1 mg ml$^{-1}$ of ampicillin or kanamycin at 37 °C until absorbance reached 0.8 at 600 nm. The recombinant proteins were expressed upon addition of 1 mM isopropyl-β-D-thiogalactoside for 16 h at 16 °C. After centrifugation, the bacterial pellet was submitted to specific purification steps[22].

Talin R1–R2–R3, R11, and R7–R8 were bound to Ni-NTA (Ni$^{2+}$-nitrilotriacetic acid)-Agarose (Macherey-Nalgene), washed with 50 mM Tris pH 7.8, 500 mM NaCl, 20 mM imidazole, 1 mM β-mercaptoethanol (BME), eluted with 50 mM Tris pH 7.8, 500 mM NaCl, 250 mM imidazole, 1 mM BME, dialyzed in 20 mM Tris pH 7.8, 100 mM KCl, 1 mM DTT, frozen in liquid nitrogen, and stored at −80 °C.

mCherry-RIAM 1-306, talin F2–F3, F2–F3–R1–R2–R3, R1–R8, and EGFP-Vh containing a N-terminal GST (Glutathione-S-transferase) tag, were bound to glutathione-Sepharose (GE Healthcare), washed with 50 mM Tris pH 7.8, 500 mM NaCl, and eluted with 50 mM Tris pH 7.8, 500 mM NaCl and 50 mM reduced L-Glutathione (Sigma-Aldrich). For mCherry-RIAM 1-306 and EGFP-Vh, GST was cleaved by PreScission protease (GE Healthcare) in 50 mM Tris pH 7.8 and 500 mM NaCl and GST was eliminated by Glutathione-Sepharose chromatography. mCherry-RIAM 1-306 and talin F2–F3, F2–F3–R1–R2–R3, and R1–R8 were then bound to Ni-NTA-Agarose, washed with 50 mM Tris pH 7.8, 500 mM NaCl, 20 mM imidazole, eluted with 50 mM Tris pH 7.8, 500 mM NaCl, 250 mM imidazole, dialyzed in 20 mM Tris pH 7.8, 100 mM KCl, 1 mM DTT, frozen in liquid nitrogen and stored at −80 °C. EGFP-Vh was further centrifuged at 300,000 × $g$ for 30 min, purified by gel filtration (Superdex 200, 16/60, GE Healthcare) in 20 mM Tris pH 7.8, dialyzed in 20 mM Tris pH 7.8, 100 mM KCl, frozen in liquid nitrogen, and stored at −80 °C.

Actin was purified from rabbit skeletal muscle acetone powder. After cycles of polymerization and depolymerization, actin was gel filtered on a Superdex G-200 column (GE Healthcare) in 5 mM Tris pH 7.8, 0.2 mM ATP, 0.1 mM CaCl$_2$, 1 mM DTT. Actin was labelled with Alexa Fluor 488, 594, and 647 Succinimidyl Ester (Invitrogen)[22]. Myosin II was extracted from rabbit skeletal muscles in a buffer containing 500 mM KCl, 100 mM K$_2$HPO$_4$. After grinding and centrifugation, the actin-containing pellet is discarded. The supernatant is submitted to cycles of precipitation in low-salt buffer, centrifugation, and resuspension in high-salt buffer. Finally, the protein was dialyzed in 20 mM KH$_2$PO$_4$/K$_2$HPO$_4$ pH 7.5, 500 mM KCl, 1 mM EDTA, and stored at −20 °C after addition of 50% glycerol.

**Sample preparation for the in vitro assay**. Micropatterning was performed by modifying an existing method as follows[53,54]. Glass coverslips (22 mm × 32 mm, Thermo Scientific/Menzel-Glaser) were first washed with milliQ water and ethanol, sonicated and irradiated for 1 min under a deep UV lamp (Ossila). The coverslips were incubated for 2 h in 0.1 mg ml$^{-1}$ PLL-g-PEG (SuSoS) dissolved in 10 mM HEPES pH 7.8 and washed with milliQ water. The chrome–quartz photomask (Toppan, France), designed with disks of 5 μm in diameter, regularly spaced by 30 μm (Fig. 1a), was cleaned by deep UV irradiation for 1 min, placed on the PLL-g-PEG-coated coverslip, and exposed to deep UV for 3 min. The chamber was made of a micropatterned coverslip attached to a glass slide (Super Frost, Thermo Scientific) with double-sided adhesive tape. The volume of a typical chamber was 50 μl. The chamber was first incubated with talin (1 μM) for 5 min at room temperature. Unbound talin was washed out with 200 μl of F-buffer (10 mM Tris pH 7.8, 25 mM KCl, 1 mM MgCl$_2$, 0.2 mM CaCl$_2$, 1 mM DTT). The surface of the disks was passivated with 100 μl of F-buffer containing 10% BSA for 5 min at room temperature and washed with 200 μl F-buffer. Finally, 100 μl of the reaction was added and the chamber was sealed with VALAP (1:1:1 mixture of vaseline, lanolin, and paraffin). A typical reaction contained: 2.4 μM actin (containing 1% Alexa647-labeled or 2% Alexa488-labeled or 2% Alexa561-labeled actin), 50 nM myosin II, 1% BSA, a salt mix (2 mM MgCl$_2$, 0.2 mM EGTA, and 25 mM KCl), and an ATP regenerating mix (2 mM ATP, 2 mM MgCl$_2$, 10 mM creatine phosphate, 3.5 U/ml creatine kinase) in G-fluo buffer (10 mM Tris pH 7.8, 0.2 mM CaCl$_2$, 0.4% methylcellulose, 5 mM DABCO and 20 mM DTT). Additional proteins such as EGFP-Vh and mCherry-RIAM 1-306 were also added. The gelsolin-capped actin filaments used in Supplementary Fig. 3 have been prepared by mixing the barbed-end capping protein gelsolin with actin filaments at a 1:600 gelsolin/actin molar ratio[22].

**Microscopy observations**. Images were acquired with a Nikon Ti Eclipse E microscope equipped with a 60X oil immersion objective (Apochromat, 1.49 NA) and coupled to a sCMOS camera (Photometrics, Prime 95B or Hamamatsu, Orca

Flash04), using the spinning disk mode (Yokogawa CSU-X1-A1) or the TIRF mode. EGFP-Vh (or Alexa488-Actin), mCherry-RIAM 1-306 (or Alexa594-actin) and Alexa647-actin were excited with 488, 561, and 642 nm lasers, respectively.

**Data analysis**. Images were acquired with MetaMorph and analyzed with ImageJ. For steady-state data, each point of the dot plots represents the mean fluorescence of a single disk (background subtracted). The bar indicates the mean. Kinetics were obtained by averaging kinetics in a large number of single disks, after background subtraction, normalization, and synchronization on the maximal value of mCherry-RIAM for each disk (Figs. 3e–g, 4e, g), only synchronization (Fig. 4d, f), or only background subtraction (Figs. 2e–g, 5d, e, and Supplementary Fig. 5b, c).

The affinity of RIAM for talin constructs was obtained by plotting the average fluorescence of RIAM in a high number of talin-coated disks as a function of the total concentration of RIAM (Fig. 1c). Since we assumed that the amount of talin in the disks was negligible compared to the total concentration of RIAM in solution, we estimated the value of the $K_d$ as the concentration of RIAM at half saturation. Note that this assumption can only underestimate the affinity.

The graphs were assembled using Igor Pro or Kaleidagraph. Statistical analysis was performed using Student $t$ test in Microsoft Excel. Experiments were reproduced 2–11 times with the same conclusions.

**Reporting summary**. Further information on research design is available in the Nature Research Reporting Summary linked to this article.

## Data availability
Data supporting this paper are available from the corresponding author upon reasonable request. A reporting summary for this article is available as a Supplementary Information file. The source data underlying Figs. 1c, 2e–g, 3e–g, 4d–g, 5d–g, and Supplementary Figs. 1b, 2, 3b, d, 4, 5b, c are provided as a Source data file. Source data are provided with this paper.

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

## Acknowledgements

This work was supported by Agence Nationale pour la Recherche Grants ANR-16-CE13-0007-02 PHAGOMECANO and ANR-18-CE13-0026-01 RECAMECA (to C.L.C.). The present work has benefited from the Light Microscopy facility of Imagerie-Gif, (http://www.i2bc.paris-saclay.fr), member of IBiSA (http://www.ibisa.net), supported by "France-BioImaging" (ANR-10-INBS-04-01), and the Labex "Saclay Plant Sciences" (ANR-10-LABX-0040-SPS). We thank Annabelle Fente for initial observations of mCherry-RIAM binding to talin, Jun Qin (Cleveland Clinic, USA) for the gift of the mouse RIAM cDNA. We thank Florence Niedergang and the members of the "Cytoskeleton Dynamics and Motility" team for helpful discussions.

## Author contributions

C.V. performed the microscopy experiments, analyzed the data, and prepared the figures. V.H. cloned cDNAs. C.V., V.H., and C.L.C. purified and characterized the proteins used in this study. C.L.C. designed the experiments, supervised the project, and wrote the paper.

## Competing interests

The authors declare no competing interests.
