## [Peer Review File · Nature Communications]

Reviewers' comments:

Reviewer #1 (Remarks to the Author):

This technically very elegant study analyzes the force-dependent interaction of the two adhesion proteins RIAM and vinculin to talin, a crucial integrin activator and force transducer. By further developing a previously established *in vitro* reconstitution assay, the authors demonstrate the expected actomyosin-dependent binding of vinculin to talin's R1–R3 domain but also provide direct evidence for a force-dependent interaction of vinculin with talin R11. The authors then show that actomyosin forces induce an unbinding of RIAM from talin R1–R3 and R11, while the interaction of RIAM with talin R7–R8 seems insensitive to actomyosin forces. Combining protein fragments from talin, vinculin and RIAM in one assay reveals a concomitant RIAM dissociation from and vinculin association with talin R1–R3 and talin R11 upon actomyosin contraction. Increasing concentrations of RIAM inhibit vinculin binding to R1–R3, but not to R11 suggesting a differential regulation of force-dependent vinculin binding to these domains.

Overall, this is a well-written manuscript, the data are clear and statistically sound, and the results provide intriguing insights into the regulation of talin by mechanical force. Previous work already indicated that RIAM and vinculin undergo mutual exclusive interaction with R1–R3 (Goult et al. JBC, 2013), which is nicely confirmed in this paper. In addition, the manuscript shows how the additional RIAM/vinculin binding sites in R7–R8 and in R11 are regulated by actomyosin force. It would be nice to see whether increasing concentrations of vinculin (i.e. higher concentrations than used in Fig. 4E, G) can displace RIAM from talin, as this has been suggested before (Li et al, MBoC, 2013). However, the current data sets are convincing and do not require, from my perspective, any major adjustments. The only minor drawback of the study is the absence of any attempts to confirm the observations in an independent fashion. It would certainly increase the impact of the study if some of the predictions were being tested in cell culture experiments. I would not consider such cell culture experiments mandatory, because the data are conclusive and deserve to be published.

Reviewer #2 (Remarks to the Author):

In this manuscript, the authors use an elegant *in vitro* reconstitution system with time-lapse imaging to investigate mechanoresponses of talin in RIAM and vinculin bindings. They clearly show that RIAM dissociates from and vinculin binds to talin domains in response to assembly of the actomyosin cytoskeleton. They observe that different talin domains show distinct responses in actomyosin-dependent RIAM dissociation and vinculin association. With kinetics analyses, they show the sequential actions of talin in which actomyosin accumulation induces RIAM dissociation that is followed by vinculin binding. Furthermore, they reveal that RIAM inhibits actomyosin-dependent binding of vinculin to talin.

This is an interesting and well-written paper that would contribute to our understanding of molecular basis for mechanoresponses of cell adhesion complexes. However, the reviewer has a few concerns that should be addressed before publication.

1) The authors assume that “force” is responsible for actomyosin-dependent RIAM dissociation from and vinculin association with talin. However, they applied actin and myosin II at a time and did not distinguish individual actions of these components, e.g., actin polymerization, F-actin bundling by myosin II, and force generation by actin-myosin II sliding. Therefore, it is currently unclear whether force is a true stimulus that induces the RIAM dissociation and the vinculin association in the system of this study. Several additional controls should be tested. For example, addition of 1) actin alone, and 2) actin together with myosin II that can crosslink F-actin but has no motor activity (e.g., N-

ethylmaleimide-treated myosin II; Smith et al. 2007 Biophys J) would provide good controls. Furthermore, effects of inhibition of myosin II ATPase activity need to be tested. Even though conventional blebbistatin may not be compatible with time-lapse fluorescence imaging, a photostable and non-fluorescent derivative of blebbistatin is now commercially available.

2) Even in the presence of actomyosin, talin R7-R8 shows no apparent vinculin binding and only moderate RIAM dissociation. The authors discuss that the poor response of R7-R8 in vinculin binding and RIAM dissociation may arise from high mechanical stability of R7-R8. However, considering the fact that R7-R8 directly binds to F-actin (Atherton et al. 2015 Nat Commun), a simple possibility is that F-actin binding to R7-R8 may block RIAM dissociation from and vinculin binding to R7-R8. Alternatively, simultaneous pulling at R13 and at R7-R8 by actomyosin through F-actin binding to R13 and R7-R8 may result in no apparent tension development between R13 and R7-R8, leading to little RIAM dissociation/vinculin association.

3) In this in vitro study, the authors use the vinculin head domain as a vinculin model and show its actomyosin-dependent binding to talin domains. On the other hand, studies using cells have reported that in contrast to full-length vinculin, the head domain only of vinculin constitutively binds to talin at focal adhesions independently of actomyosin activity (Humphries et al. 2007 J Cell Biol; Carisey et al. 2013 Curr Biol; Hirata et al. 2014 Am J Physiol Cell Physiol). Please discuss why the vinculin head domain behaves differently in talin binding between in vitro and in cells and how relevant the results of vinculin head-talin binding obtained in this study to vinculin-talin binding at focal adhesions in cells.

Point-by-point response to reviewers

The comments of the reviewers in bold (Q) are followed by our answers in italics (A).

Reviewer #1:

This technically very elegant study analyzes the force-dependent interaction of the two adhesion proteins RIAM and vinculin to talin, a crucial integrin activator and force transducer. By further developing a previously established in vitro reconstitution assay, the authors demonstrate the expected actomyosin-dependent binding of vinculin to talin's R1-R3 domain but also provide direct evidence for a force-dependent interaction of vinculin with talin R11. The authors then show that actomyosin forces induce an unbinding of RIAM from talin R1-R3 and R11, while the interaction of RIAM with talin R7-R8 seems insensitive to actomyosin forces. Combining protein fragments from talin, vinculin and RIAM in one assay reveals a concomitant RIAM dissociation from and vinculin association with talin R1-R3 and talin R11 upon actomyosin contraction. Increasing concentrations of RIAM inhibit vinculin binding to R1-R3, but not to R11 suggesting a differential regulation of force-dependent vinculin binding to these domains. Overall, this is a well-written manuscript, the data are clear and statistically sound, and the results provide intriguing insights into the regulation of talin by mechanical force. Previous work already indicated that RIAM and vinculin undergo mutual exclusive interaction with R1-R3 (Goult et al. JBC, 2013), which is nicely confirmed in this paper. In addition, the manuscript shows how the additional RIAM/vinculin binding sites in R7-R8 and in R11 are regulated by actomyosin force.

Q1. It would be nice to see whether increasing concentrations of vinculin (i.e. higher concentrations than used in Fig. 4E, G) can displace RIAM from talin, as this has been suggested before (Li et al, MBoC, 2013). However, the current data sets are convincing and do not require, from my perspective, any major adjustments.

A1. We understand that some experiments could have been more documented. We thank the reviewer for recognizing that our data are convincing and for not requesting an addition to our data set.

Q2. The only minor drawback of the study is the absence of any attempts to confirm the observations in an independent fashion. It would certainly increase the impact of the study if some of the predictions were being tested in cell culture experiments. I would not consider such cell culture experiments mandatory, because the data are conclusive and deserve to be published.

A2. Although our laboratory is fully mobilized for the reconstitution and characterization of mechanosensitive machineries, we agree that, for our next studies, we should collaborate with other groups to explore the cellular aspects of these questions. Here, our goal was to demonstrate and characterize the mechanosensitive transition from a talin-RIAM to a talin-vinculin complex, which was suggested by the work of other cell biology and structural biology groups as recalled by reviewer 1. Our study is therefore based on a preexisting physiological relevance.

Reviewer #2:

In this manuscript, the authors use an elegant in vitro reconstitution system with time-lapse imaging to investigate mechanoresponses of talin in RIAM and vinculin bindings. They clearly show that RIAM dissociates from and vinculin binds to talin domains in response to assembly of the actomyosin cytoskeleton. They observe that different talin domains show distinct responses in actomyosin-dependent RIAM dissociation and vinculin association. With kinetics analyses, they show the sequential actions of talin in which actomyosin accumulation induces RIAM dissociation that is followed by vinculin binding. Furthermore, they reveal that RIAM inhibits actomyosin-dependent binding of vinculin to talin. This is an interesting and well-written paper that would contribute to our understanding of molecular basis for mechanoresponses of cell adhesion complexes. However, the reviewer has a few concerns that should be addressed before publication.

Q1a. The authors assume that “force” is responsible for actomyosin-dependent RIAM dissociation from and vinculin association with talin. However, they applied actin and myosin II at a time and did not distinguish individual actions of these components, e.g., actin polymerization, F-actin bundling by myosin II, and force generation by actin-myosin II sliding. Therefore, it is currently unclear whether force is a true stimulus that induces the RIAM dissociation and the vinculin association in the system of this study. Several additional controls should be tested. For example, addition of 1) actin alone, and 2) actin together with myosin II that can crosslink F-actin but has no motor activity (e.g., N-ethylmaleimide-treated myosin II; Smith et al. 2007 Biophys J) would provide good controls.

A1a. It is true that the association of actin and myosin II could influence our experimental system in different ways, in addition to force application. In our first article, which described this experimental approach, several controls, similar to those requested by the reviewer, had been carried out (Ciobanasu et al 2014 Nature Communications). In particular, we used an “isotropic” actomyosin array. In these conditions, the actomyosin array is made of prepolymerized capped-filaments that do not elongate and are too short to be crosslinked by Myosin II. In these conditions actomyosin stimulated the binding of vinculin to a talin-coated surface. However, actin alone or myosin II alone did not stimulate the binding of vinculin to talin. Altogether, these results showed that 1) actin polymerization is not required, 2) the binding of F-actin alone to talin is not sufficient, 3) the presence of Myosin II alone is not sufficient, 4) actin crosslinking by Myosin II is not required, 5) myosin II is required (Figure 2 a-h, Ciobanasu et al. 2014, Nature Communications).

In the previous version of this manuscript, we compared the association of vinculin and RIAM to talin-coated disks in the presence of polymerizing actin and myosin II or without actin and myosin II. We made additional controls during this project, which were not included in the first version of our manuscript. In particular, we measured the binding of RIAM and vinculin to talin R1-R2-R3-coated disks in the presence of gelsolin-capped actin filaments + myosin II or gelsolin-capped actin filaments alone. These additional controls, which distinguish the individual action of myosin II in the association of vinculin and the dissociation of RIAM, are now included in this revised version as Figure S3. The results are described as follows:

p.5: “Using a procedure that we have already validated for full-length talin ²¹, we confirmed that the actomyosin-dependent increase in Vh-talin interaction depends on myosin II and not on the bundling and polymerization of the actin network, even for a minimal talin like R1-R2-R3 (Figure S3A, S3B).”

p.5: “Like the actomyosin-dependent binding of Vh to talin constructs, the actomyosin-dependent dissociation of RIAM from this minimal talin does not require the polymerization of actin, nor the formation of actomyosin bundles, and depends on the presence of myosin II in the assay (Figure S3C, S3D).”

We also show our new figure S3 below:

Figure S3. Myosin II promotes the binding of Vh to talin R1-R2-R3 and the dissociation of RIAM from talin R1-R2-R3. (A, C) Representative images of the recruitment of Vh (A) or RIAM (C) in disks coated with talin R1-R2-R3 in the absence (left) or presence (right) of myosin II. Conditions: 100 nM EGFP-Vh (A) or 100 nM mCherry-RIAM 1-306 (C), 4.8 μ M pre-formed gelsolin-capped short actin filaments, 25 nM myosin II, 1 μ M of talin R1-R2-R3 during the coating step. Bar = 10 μ m. (B, D) The recruitment of Vh (B) or RIAM (D) is quantified in the conditions described in (A) and (C) respectively. Each data point represents the mean fluorescence of a single disk, the bar shows the mean. n = 112 disks. ****, $P < 0.0001$ using a two-tailed t-test.

It would be interesting to test NEM-myosin that can crosslink F-actin but has no motor activity, as suggested by reviewer 2. However, it is difficult to compare NEM-myosin and active myosin because the crosslinking activity of inactive NEM-myosin is much stronger than that of active myosin II. Also, in the absence of motor activity, large bundles rarely associate with disks. Indeed, the myosin-dependent movement of actin filaments increases the probability that they bind to the talin-coated disks. The size of these bundles also limits their diffusion in the chamber. Even when they bind, these bundles cover only a small part of the surface of the talin-coated disks, which is difficult to compare with a control where actomyosin covers a large part of the surface. We have already encountered these problems when old myosin preps contained too many “dead myosins”, which are known to become strong crosslinkers without motor activity. Therefore, based on our experience, we prefer not to use NEM-myosin to test the role of F-actin bundling because the amount of F-actin bundles cannot be compared to the amount of actomyosin associated to talin.

Q1b. Furthermore, effects of inhibition of myosin II ATPase activity need to be tested. Even though conventional blebbistatin may not be compatible with time-lapse fluorescence imaging, a photostable and non-fluorescent derivative of blebbistatin is now commercially available.

A1b. We agree that inhibition of myosin II ATPase activity by blebbistatin is an important control. We carried out this control for the actomyosin-dependent interaction between full-length talin and Vh in the conditions used in this study (see figure below). We have not published these results and we no longer carry out such controls for several reasons.

The first problem, as mentioned by reviewer 2, is that illumination at 450-490 nm induces the formation of an inactive blebbistatin product that is highly fluorescent in the channel used for vinculin imaging. This photoproduct is also highly reactive with proteins, which is responsible for its known cytotoxicity. Although, as mentioned by reviewer 2, it is impossible to record a time lapse, it remains possible to take a snapshot of vinculin and actin at the end of a reaction (see figure below).

Although the photosensitivity problem of blebbistatin can be solved by using an alternative compound, as suggested by reviewer 2, our main reason for not using blebbistatin is that it inhibits actomyosin accumulation in the disks. With no actomyosin in the disks, this control containing blebbistatin does not differ much from that of omitting both actin and myosin. However, by using an appropriate concentration of blebbistatin, actomyosin self-organizes before stopping, which is comparable to the condition without blebbistatin. Although these conditions allowed us to show convincingly that myosin inhibition prevents Vh binding to disks coated with full-length talin, this transient self-organization is very difficult to reproduce quantitatively. Any attempt to add blebbistatin during the experiment, to reverse vinculin binding, creates a flow in the chamber that removes the actomyosin network. For all these reasons, we have stopped our efforts to carry out controls with blebbistatin. We show below the best results we have obtained with blebbistatin but we do not plan to add this figure to this revised manuscript.

Blebbistatin inhibits the actomyosin-dependent binding of vinculin head to full-length talin. Conditions: 100 nM EGFP-Vh, 2.4 μ M actin (2% Alexa594-labeled), 50 nM myosin, 1 μ M full-length talin during the coating step, in the absence (A) or presence of 68 μ M blebbistatin (B). Images were taken 1000 s after the reaction was added into the chamber. The images are color-coded using the fire LUT of ImageJ. Scale bar in time lapses = 10 μ m. (C) Quantification of the average fluorescence of EGFP-Vh in talin-coated disks. Each dot represents the mean fluorescence of a single disk. The bar shows the mean. $n = 16$.

Q2. Even in the presence of actomyosin, talin R7-R8 shows no apparent vinculin binding and only moderate RIAM dissociation. The authors discuss that the poor response of R7-R8 in vinculin binding and RIAM dissociation may arise from high mechanical stability of R7-R8. However, considering the fact that R7-R8 directly binds to F-actin (Atherton et al. 2015 Nat Commun), a simple possibility is that F-actin binding to R7-R8 may block RIAM dissociation from and vinculin binding to R7-R8. Alternatively, simultaneous pulling at R13 and at R7-R8 by actomyosin through F-actin binding to R13 and R7-R8 may result in no apparent tension development between R13 and R7-R8, leading to little RIAM dissociation/vinculin association.

A2. We thank the reviewer for this very interesting suggestion. We have added this alternative interpretation to the discussion of the revised version of our manuscript as follows (p.8):

“The fact that R8 is the C-terminal part of ABD2 allows alternative interpretations of the weak mechanosensitivity of R7-R8²³. Indeed, we cannot exclude that the binding of actin filaments

stabilizes R8 and prevents RIAM dissociation and vinculin association. Alternatively, if actomyosin generates a pulling force on R13 (ABD3) and R7-R8 (ABD2) simultaneously, the apparent tension between R13 and R7-R8 could be reduced, as suggested by FRET measurement in cells ⁴², leading to weak dissociation of RIAM and association of vinculin.”

Q3. In this in vitro study, the authors use the vinculin head domain as a vinculin model and show its actomyosin-dependent binding to talin domains. On the other hand, studies using cells have reported that in contrast to full-length vinculin, the head domain only of vinculin constitutively binds to talin at focal adhesions independently of actomyosin activity (Humphries et al. 2007 J Cell Biol; Carisey et al. 2013 Curr Biol; Hirata et al. 2014 Am J Physiol Cell Physiol). Please discuss why the vinculin head domain behaves differently in talin binding between in vitro and in cells and how relevant the results of vinculin head-talin binding obtained in this study to vinculin-talin binding at focal adhesions in cells.

A3. *We agree that this point needs to be clarified. From our point of view there are no contradictions between the observations made in vitro and in cells. We have added the following paragraph to explain the reasons of these apparent discrepancies in our revised manuscript (p.9):*

“Vinculin autoinhibition influences the mechanosensitivity of the talin-vinculin complex. Several biochemical, structural and cellular studies compared the recruitment of the constitutively active vinculin head (Vh) and the autoinhibited full-length vinculin (VFL) in FAs, leading to apparent discrepancies. In cells, Vh remains associated to talin in FAs after myosin inhibition by blebbistatin, whereas full-length vinculin (VFL) dissociates ^{24,47,48}. The recruitment of VFL to FAs is restored by cell stretching, demonstrating the force-dependence of the talin-vinculin interaction, whereas Vh binding is not increased ²⁴. The slow dissociation of Vh from talin after force release, observed in vitro ^{20,21}, could explain the slow dissociation of Vh in cells after blebbistatin treatment, whereas the fast dissociation of VFL could result from the reassociation of the tail to the head of vinculin in the absence of actomyosin. Indeed, actomyosin force acts on vinculin to maintain the active open conformation of vinculin ^{48,49}. The saturation of talin by Vh would explain why talin does not recruit more Vh after cell stretching. Thus, in vitro, and probably in cells, the mechanosensitivity of the talin-Vh interaction depends on Vh concentration. Because Vh is not autoinhibited like VFL, it binds to partially exposed VBSs in the least stable helical bundles of non-stretched talin, provided that Vh concentration is high enough ²¹. At the low concentration of Vh used in our past and present in vitro studies (22-100 nM), Vh mimics VFL by displaying a very weak constitutive binding, which is greatly enhanced by the application of actomyosin force to talin (Figure 2E).”

New references are (47) Humphries, et al. J. Cell Biol. (2007), (48) Carisey, et al. Curr. Biol. (2013), (49) Grashoff, et al. Nature (2010). (24) Hirata, et al. Am J Physiol Cell Physiol (2014) was already cited.

REVIEWERS' COMMENTS:

Reviewer #2 (Remarks to the Author):

The authors have satisfactorily addressed my comments.